# A11YN: Aligning LLMs for Accessible Web UI Code Generation

## Abstract

Large language models (LLMs) have recently demonstrated strong capabilities in generating functional and aesthetic web interfaces directly from instructions. However, these models often replicate accessibility flaws from their training data, resulting in interfaces that exclude users with diverse needs and contexts. To address this gap, we introduce A11yn, the first method that aligns code-generating LLMs to reliably produce accessibility-compliant web UIs. A11yn optimizes a novel reward function that penalizes violations of the Web Content Accessibility Guidelines (WCAG), with penalties scaled to the severity of each violation as identified by an accessibility testing engine. To support training, we construct UIReq-6.8K, a dataset of 6,800 diverse instructions for web UI generation. For evaluation, we introduce RealUIReq-300, a benchmark of 300 real-world web UI requests grounded and manually curated from public web pages, spanning a broad range of use cases. Empirical results show that A11yn significantly outperforms strong baselines, lowering the Inaccessibility Rate by 60% over the base model while preserving semantic fidelity and visual quality of generated UIs. These findings demonstrate that accessibility can be systematically optimized within LLMs, showing the feasibility of aligning code generation for accessibility.

## 1 Introduction

Large language models (LLMs) have opened up a new frontier in front-end development. With a simple prompt, language models can generate complete web interfaces, from static HTML pages to complex, interactive components (Zhou et al., 2025). Recent benchmarks and systems have shown that LLMs can synthesize semantically accurate and visually coherent UIs, even emulating modern design patterns and wide range of front-end frameworks (Xiao et al., 2025; Lu et al., 2025). This has fueled growing research interest in LLM-based UI generation systems, which aim to improve layout, fidelity, interactivity, and functional completeness.

However, **Accessibility** remains a critical yet underexplored dimension in LLM-based web UI development. Web accessibility is a key principle that ensures anyone, even people with disabilities to perceive and navigate web interfaces. For example, blind users rely on screen readers to interpret content, while people with limited motor control need to navigate without a mouse. For millions of people, these accommodations determine whether a website is usable or completely inaccessible. To this end, the W3C defines the Web Content Accessibility Guidelines (WCAG) to formalize accessibility standards. Yet, audits report widespread non-compliance, with over 90% of public web pages containing detectable violations (Mowar et al., 2025). These shortcomings disproportionately affect users with visual or motor impairments, reinforcing barriers to digital participation.

LLMs, trained on massive web corpora with such accessibility flaws, frequently replicate them in the generated UIs. Prior studies (Suh et al., 2025; Mowar et al., 2025; Aljedaani et al., 2024; Guriţă & Vatavu, 2025) confirm that LLMs omit key accessibility elements, such as alternative text, semantic landmarks, and properly labeled form controls, resulting in inaccessible interfaces. This raises a core research question: *Can we align LLMs to natively generate web UIs that are more accessible?*

In this work, we introduce **A11yn** (pronounced *align*), the first framework to align code LLMs for accessibility-aware web UI generation. To promote accessibility, we devise a novel reward function using accessibility violations that are detected using Axe Core (Deque Systems, 2015), a widely adopted WCAG auditing tool that reports issues across four severity levels. These violations

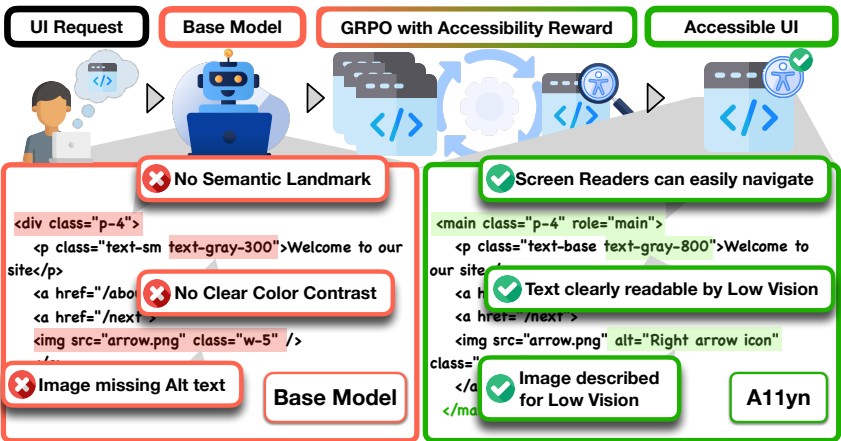

Figure 1: **A11yn enhances accessibility in UI-generative LLMs.** Whereas base models often produce inaccessible code, A11yn generates web UIs with improved accessibility features, supporting screen readers with better readability, smoother navigation, and clearer image descriptions.

are mapped to severity-weighted penalties, which are then converted into a bounded reward. The resulting reward signal is used to directly optimize the code LLM policy through Group-Relative Policy Optimization (GRPO) (Shao et al., 2024).

To support training, we construct UIReq-6.8K, an instruction-only dataset of 6,800 natural language UI generation requests spanning diverse domains and component requirements. This dataset enables reinforcement learning without relying on supervised fine-tuning data, which is difficult to collect at scale due to its scarcity and annotation cost of accessible code examples. For evaluation, we curate RealUIReq-300, a benchmark of 300 real-world web UI generation tasks, each request specified with detailed metadata such as purpose, page type, application domain, and required components. We empirically demonstrate that A11yn substantially minimizes accessibility violations, reducing Inaccessibility Rate by 60% compared to the base model, while preserving the appearance and semantic fidelity of the generated UIs. Our results suggest that accessibility can be effectively integrated as a learnable behavior within the LLM generation pipeline, bringing us closer to truly inclusive UI code generation systems.

## 2 RELATED WORK

**LLM-based UI Code Generation.** Prior research has applied specialized models to automate the translation of designs or descriptions into code. Early work like ReDraw (Moran et al., 2018) used a learned model to assemble mobile UI code from image mock-ups. With the advent of large language models (LLMs), generating UI code directly from high-level natural language descriptions has become feasible. For instance, UICoder (Wu et al., 2024) iteratively fine-tunes pre-trained LMs with SFT on a self-generated SwiftUI training dataset, that is filtered in scale with automatic compiler feedback and a CLIP-based model. On the web UI generation side, WebGen-Bench (Lu et al., 2025) provides a benchmark that is designed to evaluate LLM-based agents in generating fully functional, multi-page web applications, featuring diverse application generation instructions and automated web navigation tests to assess functionality.

**Post Training LLM for alignment.** Fine-tuning LLMs with extra objective signals has become widespread. Reinforcement Learning from Human Feedback (RLHF) (Christiano et al., 2017; Ouyang et al., 2022) adopts Proximal Policy Optimization (PPO) (Schulman et al., 2017) for LLMs to align them with human preferences. However, PPO requires training a critic alongside the policy, adding both computational overhead and engineering complexity. A recent alternative simplifies this process: Direct Preference Optimization (DPO) (Rafailov et al., 2023) reformulates preference learning by directly adjusting the model based on pairwise preferences, eliminating the need for online training. Meanwhile, Group-Relative Policy Optimization (GRPO) (Shao et al.,

2024) extends PPO by removing the critic and instead compute advantage as rewards normalized across a batch of completion samples. GRPO has shown promise in both improving performance in verifiable domains (Mathematical Association of America, 2025) as well as aligning LLMs with human values and safety constraints Li et al. (2025).

**Improving Web UI Accessibility with LLMs.** Real-world web data often contains accessibility violations, leading LLMs trained on such data to reproduce accessibility flaws in generated UI code (Martins & Duarte, 2024; Guriţă & Vatavu, 2025; Ahmed et al., 2025; Aljedaani et al., 2024). While LLMs can sometimes surpass human-written code in accessibility, they still struggle in compliance (Suh et al., 2025). Novice developers using AI assistants also frequently omit key practices, underscoring current limitations (Mowar et al., 2025). To address these issues, practical tools like CodeA11y (Mowar et al., 2025), a VS Code plugin (Calì et al., 2025), and ACCESS for real-time in-DOM correction (Huang et al., 2024) provide LLM-based accessibility support. Feeda11y (Suh et al., 2025) further improves accessibility by applying feedback loops to iteratively prompt LLMs for better compliance. Yet such methods remain costly because the inference overhead often exceeds the training cost. This motivates training models that natively generate accessible code by design.

## 3 METHODOLOGY

A11yn aligns code-generative LLMs to improve the *accessibility* of generated web UI code. The method incorporates a novel accessibility reward through reinforcement learning. Below, we outline (1) the preliminaries of the approach, (2) the reward function design, and (3) the training pipeline.

### 3.1 PRELIMINARY

GRPO (Shao et al., 2024) is a policy gradient method that simplifies PPO (Schulman et al., 2017) by removing the critic network and instead comparing sampled completions. For a prompt $q$, the policy $\pi_\theta$ samples $G$ candidate completions $\{o_1, \ldots, o_G\}$, each assigned a scalar reward $r_i$ (section 3.2). GRPO normalizes these via $\hat{A}_i = \frac{r_i - \bar{r}}{\sigma}$, emphasizing relative accessibility improvements rather than the absolute scores to stabilize updates. At the token level generation, the probability ratio $r_t^{(i)}(\theta) = \pi_\theta(o_{i,t} \mid q, o_{i,<t}) / \pi_{\theta_{\text{old}}}(o_{i,t} \mid q, o_{i,<t})$ quantifies how the new policy changes the likelihood of generating token $o_{i,t}$ conditioned on the prompt $q$ and previously generated tokens $o_{i,<t}$. The clipped surrogate loss $L_t^{(i)}(\theta)$, then bounds large ratios to avoid overcorrection. The overall objective averages these token-level terms with a KL penalty against a frozen reference policy $\pi_{\text{ref}}$:

$$J_{\text{GRPO}}(\theta) = \mathbb{E}_{q, \{o_i\} \sim \pi_{\theta_{\text{old}}}} \left[ \frac{1}{G} \sum_{i=1}^{G} \frac{1}{|o_i|} \sum_{t=1}^{|o_i|} \underbrace{\min\left( r_t^{(i)}(\theta) \hat{A}_i, \; \text{clip}\left( r_t^{(i)}(\theta), \, 1 - \epsilon, \, 1 + \epsilon \right) \hat{A}_i \right)}_{L_t^{(i)}(\theta)} \right]$$

$$- \beta \, D_{\text{KL}}(\pi_\theta \| \pi_{\text{ref}}).$$

$$(1)$$

The KL penalty constrains excessive divergence from the reference policy, preserving general code generation ability while guiding completions toward accessibility. In our task, GRPO is especially well-suited due to its data-efficient and stable optimization method, whereas SFT relies on large paired accessible code datasets and PPO entails an engineering and computation overhead of training a critic model that estimates value for accessibility.

### 3.2 ACCESSIBILITY REWARD

To guide the A11yn policy towards generating accessible web UI code, we design a reward function with Web Content Accessibility Guidelines (WCAG) auditing tool. After current policy model $\pi_\theta$ generating each web UI code output, we run Axe-core (Deque Systems, 2015), a widely used open-source accessibility engine, to detect violations of the WCAG. For each response completion, Axe-core returns a list of affected DOM nodes, where each violation is classified by severity $v \in \{\text{Minor, Moderate, Serious, Critical}\}$. For a UI output $o_i$, we let $V(o_i)$ denote the set of severity

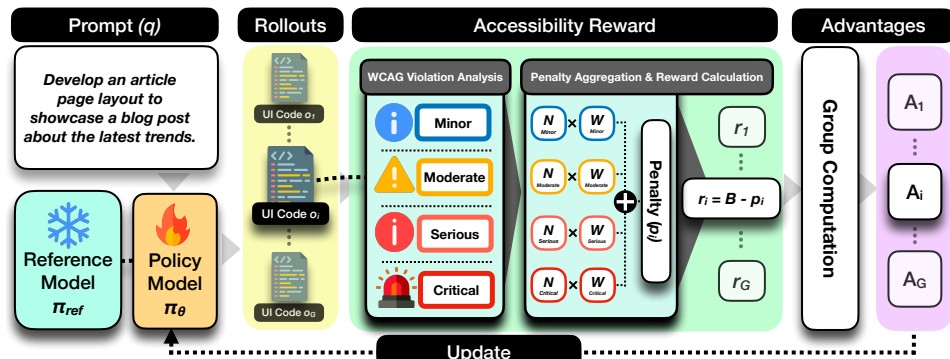

Figure 2: **A11yn optimizes accessibility through reinforcement learning (GRPO).** For an instruction $q$, the policy LLM $\pi_\theta$ generates candidate UI codes $\{o_1, \dots, o_G\}$. Each code receives an accessibility reward $\{r_1, \dots, r_G\}$, which is normalized within the set of candidates to compute advantages. The policy $\pi_\theta$ is then updated via policy gradient using these advantages.

levels detected in that output. The number of nodes associated with each violation level is counted and denoted as $N_v$. The total penalty $p_i$ for a UI output $o_i$ is computed by aggregating the affected DOM nodes, weighting each by its severity:

$$p_i = \sum_{v \in \mathcal{V}(o_i)} N_v \cdot w_v \tag{2}$$

where severity weights are $w_v \in \{0.1, 0.2, 0.3, 0.4\}$ corresponding respectively to Minor, Moderate, Serious, and Critical violations. While the scales are arbitrary, the scheme ensures more severe violations to incur systematically larger penalties. We then convert this into a bounded reward by subtracting the penalty from a base score $B$, where we use $B = 2.0$ empirically, and clip the reward to zero for negative values.

$$r_i = B - p_i \tag{3}$$

Under this quantitative reward signaling scheme, a violation-free output converges toward $r_i \approx B$, and each violation proportionally lowers the reward. By assigning larger negative weights to more severe issues, the policy is encouraged to eliminate severe failures first. In practice, the policy model receives a solid numerical score that reflects the accessibility testing environment that is appropriate for giving RL feedback.

### 3.3    TRAINING PIPELINE

We instantiate **A11yn** as a GRPO based reinforcement learning pipeline with *rollout-reward–update* cycle that repeatedly steers the policy toward accessibility-compliant code as illustrated in fig. 2. We use Qwen2.5-Coder-7B-Instruct (Hui et al., 2024) as our policy model $\pi_\theta$ and simultaneously use its frozen copy as the reference model $\pi_{ref}$, since it is pre-trained and capable of generating web contents based on natural language request. In each iteration, a textual UI request prompt $q$ from training prompt set (section 4.1) is retrieved. The current policy $\pi_\theta$ then generates a group of $G$ candidate completions $\{o_i\}_{i=1}^{G}$, producing diverse web UI code alternatives for the same prompt. Each completion is evaluated with Axe-core (Deque Systems, 2015), where generated web contents are rendered in a headless Chromium instance and analyzed for WCAG violations. The detected violations are converted into scalar penalties using the severity-weighted mapping described in section 3.2, yielding an accessibility reward $r_i$ for each completion. Group statistics, mean $\bar{r}$ and standard deviation $\sigma$ are computed to form normalized advantages. Then, these group-normalized advantages focus updates on relative improvements among the sampled completions, favoring code patterns that have minimal WCAG violations in the same group.

## 4    DATA

### 4.1    TRAINING: UIREQ-6.8K

To train A11yn, we construct UIReq-6.8K, a reinforcement learning training dataset of 6,800 UI generation instructions. As shown in fig. 3, the dataset spans a wide range of domains and interaction patterns, supporting broad coverage of instruction types. Unlike supervised datasets, UIReq-6.8K does not fix target UIs for each request, which enables exploration and reward optimization without imposing stylistic bias. Each instruction prompt in UIReq-6.8K describes a desired user interface in natural language, specifying page type, application domain, specific web UI components, or stylistic intent (e.g. a dark-themed login screen with email and password inputs). The instruction prompts are generated using GPT-4o-mini (OpenAI et al., 2024) and guided to reflect diversity and semantic richness. Diversity is

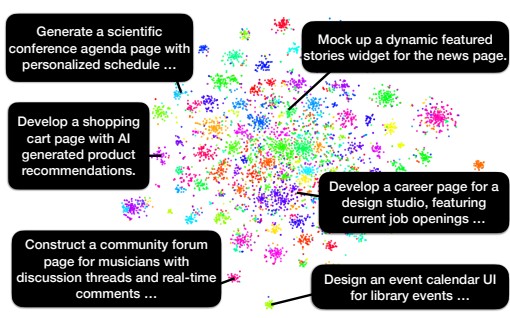

Figure 3: **t-SNE visualization of the training set.** Each point represents a UI request, with colors indicating distinct application categories. The spread shows coverage across multiple domains with examples in the figure.

achieved by covering 68 application categories (appendix A). Semantic richness is enforced through detailed requirements in instruction prompt synthesis, where every request specifies its page type, application domain, specific web UI components, and stylistic intent.

## 4.2 EVALUATION: REALUIREQ-300

| Dataset | Query Style | UI Intent Coverage | Component Details | UI Source | Query Length | Number of Queries |
|---|---|---|---|---|---|---|
| Screen2Words (Wang et al., 2021) | Single sentence, Taxonomic | ✗ | ✗ | RICO dataset (Android UI) | 6 words | 112k |
| **RealUIReq-300 (Ours)** | Multi-sentence, Request-oriented | ✓ | ✓ | Real-World Web UI | 87 words | 300 |

Table 1: **Comparison of REALUIREQ-300 with Screen2Words.** REALUIREQ-300 provides multi-sentence requests with structured intent, detailed UI component specification, and realistic phrasing grounded in real-world web UIs.

We assess the accessibility of web UI within the broader scope of natural language to web UI code generation flow. To achieve this, a realistic request-style benchmark dataset was required, one that could capture authentic user intents and interface specifications instead of relying on fully synthetic or overly simplified captions.

Screen2Words (Wang et al., 2021) is the most widely adopted dataset for natural language description of user interfaces. Built on the RICO dataset (Deka et al., 2017) of Android application UIs, its primary objective is to provide concise textual summaries of the UI screen to bridge user interfaces and natural language. While valuable in scale, the descriptions are short and taxonomic (e.g. sign in page of a social app, page displaying data status) rather than being detailed and request-oriented. Evaluating with short summaries risks emphasizing superficial matches over true task alignment. Such an absence of explicit intent or evaluation points (e.g. UI component details or requirements) in the descriptions further introduces ambiguity, making the benchmark less reliable.

To address these limitations, we introduce RealUIReq-300, a benchmark of 300 web UI requests inversely generated from manually collected webpage screenshots. As shown in fig. 4, each example was curated through a multi-stage pipeline involving screenshot collection, metadata extraction, and request generation, with GPT-4.1 (OpenAI et al., 2024) assisting in extraction and request phrasing. All metadata and requests were manually refined by the authors to correct for truncation, vague language, or missing context. This process ensured that the final requests faithfully represent the semantics of original UIs while maintaining natural and realistic phrasing. As compared in table 1, RealUIReq-300 offers multi-sentence, request-style instructions with intent, page type, UI components, and domain context specifications. This makes the evaluation set semantically rich, structurally aligned for assessing natural language to UI generation.

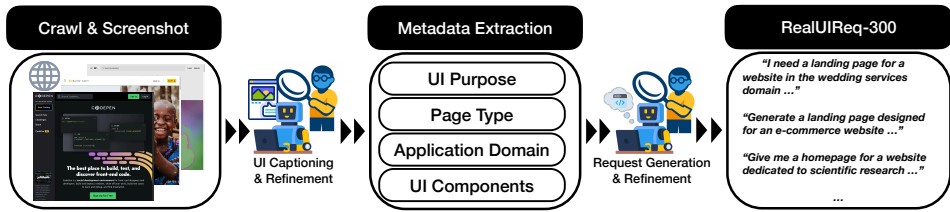

Figure 4: **RealUIReq-300 is curated from real web UIs with diverse use-case domains.** User requests are inversely generated from screenshots and metadata extracted, then refined to produce realistic instructions aligned with the original UIs.

## 5 EXPERIMENTS

### 5.1 SETUP

Our objective is to evaluate A11yn and baseline LLMs in generating web UIs with minimal accessibility violations. We further evaluate if A11yn and selected baselines are able to balance accessibility and semantic alignment with visual appeal in section 6.3.2. Each model is tested on RealUIReq-300 benchmark, an evaluation set of 300 web UI request prompts, designed to ensure consistent and controlled comparisons. Inference of all model candidates is performed with a temperature of 0.1 for near-deterministic reproducibility.

### 5.2 METRICS

To assess the accessibility of model-generated responses, we adopt a comprehensive set of evaluation metrics informed by the principles of the Web Content Accessibility Guidelines (WCAG) from the accessibility auditing tool. Our evaluation comprises three main metrics designed for robustness and fairness. First, we measure the average DOM counts with accessibility violations detected across the evaluation set, categorized by severity: **Minor**, **Moderate**, **Serious**, and **Critical**. Each severity level reflects the impact of the violation on user experience, ranging from minor impact issues to critical barriers that significantly hinder accessibility.

To account for the varying severity of accessibility violations, we propose the **Weighted Violation Score (WVS)**, which quantifies accessibility violations by assigning severity-based weights to affected DOM nodes at each severity category. The WVS is formally defined as:

$$\text{WVS} = \lambda_{\text{Minor}} \cdot N_{\text{Minor}} + \lambda_{\text{Moderate}} \cdot N_{\text{Moderate}} + \lambda_{\text{Serious}} \cdot N_{\text{Serious}} + \lambda_{\text{Critical}} \cdot N_{\text{Critical}} \quad (4)$$

where $N_{\text{Minor}}$, $N_{\text{Moderate}}$, $N_{\text{Serious}}$, and $N_{\text{Critical}}$ represent the number of violated DOM counts at each severity level from the generated code with RealUIReq-300 request prompts. The corresponding weights $\lambda$ reflect the relative impact of each category, with values of 1 for *Minor*, 2 for *Moderate*, 3 for *Serious*, and 4 for *Critical*. This formulation provides a single interpretable metric that captures both the frequency and severity of accessibility issues.

Finally, since models differ in scale and generate varying length of web contents, we adopt a normalized metric to enable fair comparison across models. Inspired by the **Inaccessibility Rate** introduced in Feeda11y (Suh et al., 2025), we calculate the ratio of weighted violations to the total number of DOM elements produced when prompted with RealUIReq-300 requests. Given our use of a different auditing tool (Axe core Deque Systems (2015)), we adapt the original formulation to incorporate the WVS, resulting in the following metric:

$$\text{Inaccessibility Rate} = \frac{\text{WVS}}{\text{No. of Total DOM Elements}} \quad (5)$$

This metric captures the normalized, severity-adjusted density of accessibility violations, allowing us to evaluate the true accessibility in proportion to UI complexity.

### 5.3 BASELINES

We compare our work against five baseline models to evaluate its relative performance. **(1) Qwen2.5-Coder-7B-Instruct** serves as the base model from which A11yn is GRPO-tuned. It re-

| Model | Average Violated DOM Counts ($\downarrow$) | | | | WVS ($\downarrow$) | IR ($\downarrow$) |
|---|---|---|---|---|---|---|
| | Minor | Moderate | Serious | Critical | | |
| Qwen2.5-Coder-7B-Instruct | 1 ($\pm$1) | 1149 ($\pm$36) | 978 ($\pm$48) | 40 ($\pm$0) | 5392 ($\pm$35) | 0.38 ($\pm$0.0) |
| + Feeda11y | 11 ($\pm$1) | 461 ($\pm$37) | 841 ($\pm$33) | 30 ($\pm$5) | 3576 ($\pm$64) | 0.21 ($\pm$0.0) |
| Qwen2.5-Coder-14B-Instruct | 3 ($\pm$2) | 846 ($\pm$35) | 1491 ($\pm$32) | 49 ($\pm$8) | 6365 ($\pm$158) | 0.43 ($\pm$0.0) |
| GPT-4.1 | 45 ($\pm$5) | 1925 ($\pm$34) | 1424 ($\pm$21) | 105 ($\pm$7) | 8588 ($\pm$100) | 0.27 ($\pm$0.0) |
| Claude Sonnet 4 | 2 ($\pm$3) | 3388 ($\pm$81) | 1435 ($\pm$14) | 282 ($\pm$10) | 12210 ($\pm$117) | 0.29 ($\pm$0.0) |
| **A11yn (Ours)** | **0** ($\pm$0) | **231** ($\pm$16) | **481** ($\pm$23) | **24** ($\pm$3) | **1918** ($\pm$65) | **0.15** ($\pm$0.0) |

Table 2: **Accessibility measures across models.** We report Average Violated DOM Counts at different severity levels. Weighted Violation Score (WVS) and Inaccessibility Rate (IR) provide severity-adjusted and normalized aggregate measures, respectively. Lower values indicate better performance. Best results are shown in **bold**, and second-best in underline.

flects the model's raw web UI code generation capability in zero shot setting without any explicit accessibility optimization. **(2) Qwen2.5-Coder-7B-Instruct** *(+Feeda11y)* is used to examine the impact of accessibility-aware prompting. This variant incorporates Feeda11y (Suh et al., 2025) prompts using a three-step iterative ReAct prompting (Yao et al., 2023) method with violation report feedbacks. **(3) Qwen2.5-Coder-14B-Instruct** is included to assess the effect of model scaling, offering a larger alternative from the same model family. In addition, we evaluate two frontier models: **(4) GPT-4.1** (OpenAI et al., 2024) and **(5) Claude Sonnet 4** (Anthropic, 2025), both of which represent the well performing models in general-purpose code and web UI generation.

## 6 RESULTS

### 6.1 QUANTITATIVE RESULTS

Table 2 summarizes the accessibility performance of A11yn against five baselines. Frontier models like GPT-4.1 and Claude Sonnet 4 yield relatively high inaccessibility rates (0.27 and 0.29), indicating that strong models do not guarantee accessible outputs. A11yn achieves the lowest Weighted Violation Score (WVS) and Inaccessibility Rate (IR), significantly outperforming both prompt-based approaches and frontier models. Compared to the base model, A11yn has critical violations reduced from 40 to 24 (40% $\downarrow$), serious from 978 to 481 (50.8% $\downarrow$), moderate from 1149 to 231 (79.9% $\downarrow$), Weighted Violation Score from 5392 to 1918 (64.4% $\downarrow$), and Inaccessibility Rate from 0.38 to 0.15 (60.5% $\downarrow$), demonstrating substantial improvements in accessibility conformity. Base model with Feeda11y shows notable improvement, achieving a WVS of 3576 and an Inaccessibility Rate of 0.21, yet remaining behind A11yn. Also, its iterative prompting brings up computational overhead, averaging 4584 intermediate tokens per request.

### 6.2 QUALITATIVE EXAMPLES

Figure 5 illustrates the rendered UI and the corresponding accessibility tree pairs for the base model and A11yn. Beyond the visible improvements, the accessibility tree provides a structural view of how assistive technologies like screen readers or switch devices interpret and navigate the UI page.

The base model (left) contains several violations across different severity levels, which are reflected directly in its accessibility tree. The weak color contrast (Serious) issue of the text, visible in the navigation bar, prevents low vision users from reliably perceiving text. The missing landmark semantics (moderate) issue leaves sections in the middle like "Featured Movies & Shows" to be stated without clear structural roles, disrupting the navigation for screen-reader users. The iframe elements with the absence of accessible names cause media contents like embedded videos to be exposed without meaningful descriptions for non-sighted users. By contrast, A11yn (right) produces UI output with strong color contrast, sections well positioned in Header–Main–Footer landmarks, and iframe elements annotated with descriptive accessible names. Its accessibility tree reflects these improvements through a coherent hierarchy with properly assigned semantic roles, demonstrating that A11yn enhances both the visual presentation and the assistive-technology interpretation of the web UI.

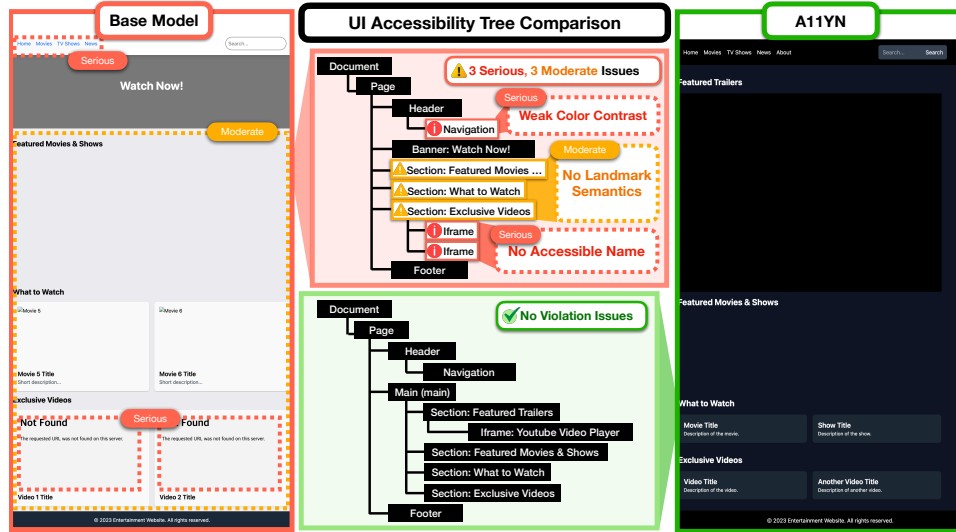

Figure 5: **Comparison of the base model (left) and A11yn (right) using both rendered UIs and their corresponding accessibility trees (middle).** The accessibility tree reveals the base model's weak color contrast, missing landmarks, and unlabeled media nodes, while A11yn provides clear landmarks, stronger contrast, and descriptive accessible names, resulting in a clean, fully interpretable structure.

## 6.3 ANALYSIS

### 6.3.1 ACCESSIBILITY IMPROVEMENT OF A11YN OVER BASELINES

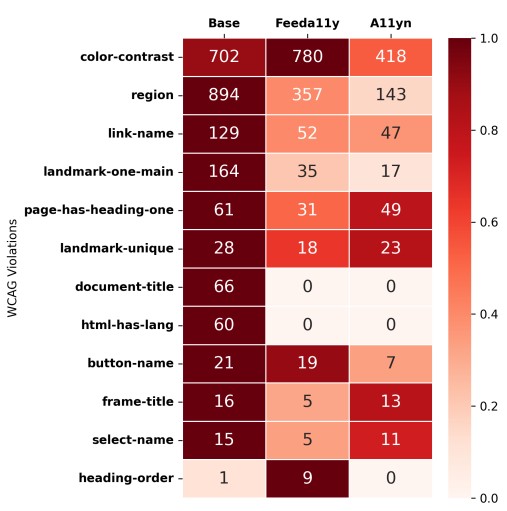

Figure 6: **Per-category violation counts across models.** Colors are normalized per row for visualization. (lighter = fewer violations, darker = more violations).

Figure 6 demonstrates a comparison of average accessibility violation distribution among the Base model, Feeda11y and A11yn across multiple violation types in rendering of the evaluation set UI requests. It reveals that A11yn reduces a range of key accessibility issues, and the most notable improvements are observed in the following violation categories:

**Region - All page content must be contained by landmarks** A11yn reduces average Region violations from 894 to 143. These violations indicate failures to encapsulate page content within landmark regions like `<main>`, `<nav>`, `<header>` tags. Proper use of HTML5 and ARIA landmarks is critical for screen reader users, allowing them to move directly to key sections of a webpage, facilitating efficient content navigation. It shows how A11yn has improved structural accessibility of the generated web UIs.

**Color Contrast - Elements must meet minimum color contrast ratio thresholds** As demonstrated in fig. 5, weak color contrast relates to insufficient contrast between text and its background. This poses a major barrier for users with low vision. WCAG recommends a minimum contrast ratio of 4.50:1. A11yn reduces the count from 702 to 418, showing enhancement in awareness of visual accessibility standards.

**Landmark-one-main - document should have one main landmark** To enhance the browsing experience for screen readers, Web UI design must allow quick and easy identification and navigation to the page's main content. With such aim, each page must include a single `<main>` landmark to clearly designate the primary content area. Using multiple or omitting the `<main>` tag can cause confusion for assistive technologies. A11yn enforces this guideline, reducing the count from 164 to 17, ensuring interpretable content hierarchies.

**Link-name - Links must have discernible text** Every hyperlink should have a clear, descriptive label to guide screen readers to understand its destination or action. Common issues include empty anchor tags or overly generic text like "click here." A reduction from 129 to 47 violations suggests A11yn reliably assigns accessible and descriptive link texts, mitigating issues like empty or duplicated links that confuse users.

### 6.3.2 ACCESSIBILITY WITH SEMANTIC ACCURACY AND AESTHETICS OF USER INTERFACES

Web UI generation task is multi-dimensional, where diverse design objectives must be considered. Effective user interfaces must guarantee accessibility while preserving semantic fidelity and visually appealing designs. Prior studies highlight that accessibility and aesthetics are often perceived in tension (Anthony, 2019), yet must be balanced rather than being treated as opposing forces (Kurosu & Kashimura, 1995; Mbipom & Harper, 2011; Le-Cong et al., 2021). Therefore, while our work primarily focuses on accessibility enhancement, it is equally important that improvements do not compromise semantic fidelity or aesthetics. To this end, we additionally evaluate appearance quality to verify whether accessibility gains are achieved without harming other key dimensions.

For this evaluation, we adopt the Appearance Score from WebGen-Bench (Lu et al., 2025), a 5-point Likert scale rated by GPT-4.1 on rendering quality, content relevance, layout harmony, and modernity. The Appearance Score serves as a core metric capturing both semantic fidelity and aesthetics. As shown in table 3, A11yn achieves the lowest inaccessibility rate, marking a 60.5% reduction over the base model, while maintaining an appearance score of 3.6. This demonstrates that A11yn substantially improves accessibility while preserving aesthetics and fidelity, achieving a balanced outcome.

| Model | Inaccessibility Rate (↓) | Appearance Score (↑) |
|---|---|---|
| Base model | 0.38 | 3.6/5 |
| Feeda11y | 0.21 | **3.7/5** |
| **A11yn (Ours)** | **0.15** | 3.6/5 |

Table 3: Comparison of models in terms of Accessibility (Inaccessibility Rate) alongside Semantic Fidelity and Aesthetics (Appearance Score on a 5-point Likert scale).

By comparison, Feeda11y achieves a higher appearance score (3.7) but retains a relatively high inaccessibility rate (0.21). This indicates that Feeda11y's improvements incidentally enhance visual quality rather than systematically addressing accessibility, reflecting a shifted emphasis. In contrast, A11yn achieves lower inaccessibility rate (0.15) while maintaining the Appearance Score intact (3.6), offering stronger evidence of accessibility enhancement with balance. Moreover, A11yn attains such outcome in a single forward pass, whereas Feeda11y relies on iterative prompting.

## 7 CONCLUSION

Web accessibility is not merely a design preference but a foundational requirement for equitable digital access. While prior efforts have explored ways to support accessibility in LLM-based code generation through prompting, feedback loops or IDE-based assistance, they remained external or computationally intensive. Furthermore, such works concluded with future calls for the need of training LLMs to inherently generate accessible web UI code. Through introducing A11yn, we take a complementary but novel path by empirically suggesting that accessibility can be systematically optimized in code-generating LLMs through post-training with reward-driven alignment. Looking ahead, we believe this paradigm can be extended beyond web UI code and into broader human-computer interaction systems such as mobile applications, AR/VR environments, and multimodal interaction platforms.

## ETHICS STATEMENT

This work aims to improve digital equity by aligning code-generating LLMs to produce accessibility-compliant web UIs, thereby reducing barriers for users with disabilities. All training data were synthetically generated through controlled prompting, and evaluation data were curated from publicly available web pages with manual refinement with sensitive or personally identifiable content removed. No human subjects or private data were involved. While misuse could enable mass generation of low-quality web pages, we mitigate this risk by committing to open release of data, code, and documentation to guide responsible, accessibility-focused research.

## REPRODUCIBILITY STATEMENT

We ensure reproducibility by documenting all datasets, training details, and evaluation procedures. Training data (UIReq-6.8K) and the evaluation benchmark (RealUIReq-300) are fully described, along with synthesis prompts. Model training used *Qwen2.5-Coder-7B-Instruct* with Group-Relative Policy Optimization, with detailed hyperparameters and hardware setups provided in appendix C. Accessibility compliance was measured using the open-source axe-core engine (Deque Systems, 2015), which is distributed under the Mozilla Public License 2.0, to detect WCAG violations and compute reward signals. Upon acceptance, we will release all code, data, and model configurations to allow independent verification and replication of results.

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

## A  UIReq-6.8K Application Domains

The training dataset covers 68 diverse application categories. Examples include:

- Business & Enterprise
- Health & Wellness
- Education & E-Learning
- Data & Analytics
- Communication & Social
- E-Commerce & Retail
- Finance & FinTech
- Real Estate & Property
- Media & Entertainment
- Food & Beverage
- Travel & Hospitality
- Developer Tools & Technology
- Science & Research
- Legal & Compliance
- Automotive & Mobility
- Government & Public Services
- Environment & Sustainability
- Security & Identity
- Non-Profit & Social Impact
- AI & Machine Learning
- Books & Reference
- Comics
- Dating
- Entertainment
- Events
- Finance
- Food & Drink
- Health & Fitness
- House & Home
- Libraries & Demo
- Lifestyle
- Maps & Navigation
- Medical
- Music & Audio
- News & Magazines
- Parenting
- Personalization
- Photography
- Productivity
- Shopping
- Social
- Sports
- Tools
- Travel & Local
- Video Players & Editors
- Weather
- Auto & Vehicles
- Beauty
- Art & Design
- Board
- Card
- Casino
- Casual
- Educational (Games)
- Music (Games)
- Puzzle
- Racing
- Role Playing
- Simulation
- Sports (Games)
- Strategy
- Trivia
- Word
- Augmented Reality
- Developer Tools
- Magazines & Newspapers
- Utilities
- Graphics & Design

## B  RealUIReq-300 Domain Distribution

| Domain | Count | Domain | Count |
|---|---|---|---|
| Education | 30 | Technology/Software | 23 |
| E-commerce | 19 | News / Media | 16 |
| Non-profit / Humanitarian | 13 | Software development | 12 |
| Social media | 12 | Tourism | 10 |
| Web development | 10 | Software as a Service (SaaS) | 9 |
| Entertainment / Media | 9 | Entertainment ticketing | 8 |
| Business consulting | 6 | Professional services | 6 |
| Academic publishing | 6 | Food / Cooking | 5 |
| Personal development | 5 | Travel / Lifestyle | 5 |
| Finance / Investing | 4 | Tech product discovery and launch platform | 4 |
| Government services | 4 | Academic research | 4 |
| Environmental activism | 4 | Health and wellness | 4 |
| Community forum | 4 | Art and design | 4 |
| B2B marketing / Business services | 4 | Creative/Portfolio | 3 |
| Business communication and collaboration | 3 | Content publishing platform | 3 |
| Medical research | 3 | Personal blog / Thought leadership | 3 |
| Financial Services | 3 | Technology / Consumer Electronics | 2 |
| Crowdfunding | 2 | Outdoor recreation / Adventure sports | 2 |
| Music streaming and sharing | 2 | Education/Interactive media | 2 |
| Wedding Planning | 2 | Events and entertainment | 2 |
| Creative Networking/Showcase | 2 | Film and photography | 2 |
| Parenting / Community | 2 | Job board / Recruitment | 2 |
| Advocacy/Activism | 2 | Healthcare technology | 2 |
| Cloud computing / AI development | 2 | Retail / Fashion | 1 |
| Search engine | 1 | Social media for developers | 1 |
| Culture/History | 1 | Community/Women's empowerment | 1 |
| Freelance marketplace | 1 | Education/Science museum | 1 |
| Service industry (barbershop) | 1 | Gaming / Media | 1 |
| Microfinance / Social impact | 1 | Cloud storage and collaboration | 1 |
| Space exploration | 1 | Humor/Lifestyle | 1 |
| Religious/Charitable Organization | 1 | | |

Table 4: 61 Domain Distribution of RealUIReq-300 dataset

## C  TRAINING CONFIGURATION

We trained *Qwen/Qwen2.5-Coder-7B-Instruct* on 8 NVIDIA A6000 GPUs (48GB VRAM each) using GRPO with vLLM-based sampling and reward modeling. Training was conducted in bfloat16 mixed-precision with gradient checkpointing enabled. Each prompt was expanded into $G = 6$ sampled completions, with a per-device batch size of 2 and gradient accumulation set to 6. To stabilize optimization, KL divergence regularization was applied with $\beta = 0.001$. Below is a summary of the key configurations used; for full details, please refer to the provided training scripts and configuration files.

| Component | Configuration |
|---|---|
| Framework | HF Transformers, TRL (GRPOTrainer), Accelerate, DeepSpeed |
| Learning rate | $5 \times 10^{-5}$ |
| Optimizer | Adam ($\beta_1 = 0.9, \beta_2 = 0.99, \epsilon = 10^{-8}$) |
| Weight decay | 0.1 |
| LR Scheduler | Cosine (warmup ratio 0.01) |
| Gradient accumulation | 6 |
| Batch size | 96 (2 per device $\times$ 8 processes $\times$ accumulation 6) |
| Epochs | 2 |
| Gradient checkpointing | Enabled |
| Precision | bfloat16 |
| Loss type | grpo |
| KL $\beta$ value | 0.001 |
| use_peft | Enabled |

Table 5: Optimization and training parameters.

| Parameter | Value |
|---|---|
| Engine | vLLM (colocated mode) |
| GPU memory utilization | 60% |
| Min-p | 0.1 |
| Top-p | 1.0 |
| Top-k | -1 |
| Temperature | 0.7 |
| Repetition penalty | 1.1 |
| Stop sequence | `</answer>` |
| Max tokens | 3072 |
| Number of Generations | 6 |
| Seed | 3407 |

Table 6: GRPO sampling parameters.

## D  ICON ATTRIBUTION

Icons in the figures are sourced from Flaticon (`https://www.flaticon.com`) and are credited to their respective creators in accordance with Flaticon's licensing requirements.

## E  USE OF LLM

We employed a large language model (LLM) to enhance the clarity and accuracy of our writing, particularly in identifying and correcting grammatical errors, typographical mistakes, and in rephrasing sentences for improved readability. Furthermore, the LLM was utilized in the data generation process to provide supplementary material in support of our study.

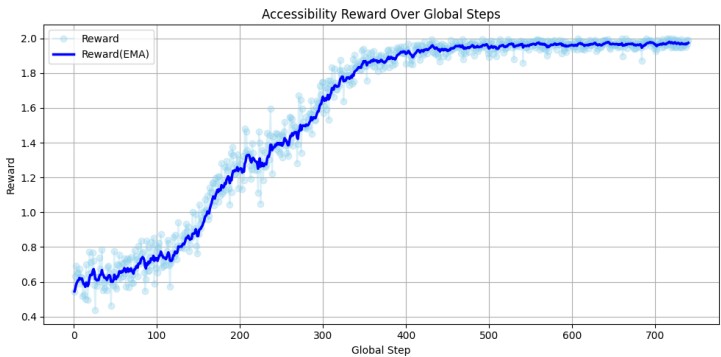

Figure 7: **Accessibility reward curve throughout training**, showing a steady increase that indicates reduced WCAG violation occurrences over time.

## F  WCAG VIOLATIONS

Below is the full list of WCAG violations.

Table 7: WCAG 2.0 — ARIA(Accessible Rich Internet Applications) Rules

| Rule ID | Impact | Description |
| --- | --- | --- |
| aria-allowed-attr | Serious, Critical | ARIA attributes are allowed for an element's role |
| aria-command-name | Serious | Every ARIA button, link, and menuitem has an accessible name |
| aria-hidden-body | Critical | `aria-hidden='true'` is not present on the `<body>` element |
| aria-hidden-focus | Serious | `aria-hidden` elements are not focusable nor contain focusable elements |
| aria-input-field-name | Moderate, Serious | Every ARIA input field has an accessible name |
| aria-meter-name | Serious | Every ARIA meter node has an accessible name |
| aria-progressbar-name | Serious | Every ARIA progressbar node has an accessible name |
| aria-required-attr | Critical | Elements with ARIA roles have all required ARIA attributes |
| aria-required-children | Critical | Elements with ARIA roles that require child roles contain them |
| aria-required-parent | Critical | Elements with ARIA roles that require parent roles are contained by them |
| aria-roles | Minor–Critical | Elements with a `role` use a valid value |
| aria-toggle-field-name | Moderate, Serious | Every ARIA toggle field has an accessible name |
| aria-tooltip-name | Serious | Every ARIA tooltip node has an accessible name |
| aria-valid-attr-value | Serious, Critical | All ARIA attributes have valid values |
| aria-valid-attr | Critical | Attributes beginning with `aria-` are valid ARIA attributes |

Table 8: WCAG 2.0 — Text Alternatives & Captions

| Rule ID | Impact | Description |
| --- | --- | --- |
| area-alt | Critical | `<area>` elements of image maps have alternate text |
| image-alt | Critical | `` has alt text or role of none/presentation |
| input-image-alt | Critical | `<input type=image>` has alternate text |
| object-alt | Serious | `<object>` elements have alternate text |
| role-img-alt | Serious | `role="img"` elements have alternate text |
| svg-img-alt | Serious | `<svg>` with img/graphics roles have accessible text |
| video-caption | Critical | `<video>` elements have captions |

Table 9: WCAG 2.0 — Keyboard, Focus & Navigation

| Rule ID | Impact | Description |
|---|---|---|
| bypass | Serious | Page has a mechanism to bypass navigation |
| nested-interactive | Serious | Interactive controls are not nested |
| scrollable-region-focusable | Serious | Scrollable content regions are keyboard accessible |
| server-side-image-map | Minor | Server-side image maps are not used |

Table 10: WCAG 2.0 — Frames & Embeds

| Rule ID | Impact | Description |
|---|---|---|
| frame-focusable-content | Serious | `<frame>`/`<iframe>` with focusable content do not have `tabindex=-1` |
| frame-title-unique | Serious | `<iframe>`/`<frame>` contain a unique `title` attribute |
| frame-title | Serious | `<iframe>`/`<frame>` have an accessible name |

Table 11: WCAG 2.0 — Forms & Names

| Rule ID | Impact | Description |
|---|---|---|
| button-name | Critical | Buttons have discernible text |
| input-button-name | Critical | Input buttons have discernible text |
| label | Minor–Critical | Every form element has a label |
| select-name | Minor–Critical | `<select>` has an accessible name |
| form-field-multiple-labels | Moderate | Form field does not have multiple label elements |

Table 12: WCAG 2.0 — Structure & Semantics

| Rule ID | Impact | Description |
|---|---|---|
| definition-list | Serious | `<dl>` elements are structured correctly |
| dlitem | Serious | `<dt>` and `<dd>` are contained by a `<dl>` |
| list | Serious | Lists are structured correctly |
| listitem | Serious | `<li>` elements are used semantically |
| document-title | Serious | Each HTML document contains a non-empty `<title>` |

Table 13: WCAG 2.0 — Parsing & Uniqueness

| Rule ID | Impact | Description |
|---|---|---|
| duplicate-id-active | Serious | Every `id` of active elements is unique |
| duplicate-id-aria | Critical | Every `id` used in ARIA and in labels is unique |
| duplicate-id | Minor | Every `id` attribute value is unique |

Table 14: WCAG 2.0 — Color & Visual Presentation

| Rule ID | Impact | Description |
|---|---|---|
| color-contrast | Serious | Foreground/background colors meet WCAG 2 AA contrast thresholds |
| link-in-text-block | Serious | Links are distinguishable from surrounding text without relying on color |
| meta-viewport | Critical | `<meta name="viewport">` does not disable text scaling and zooming |
| blink | Serious | `<blink>` elements are not used |
| marquee | Serious | `<marquee>` elements are not used |
| link-name | Serious | Links have discernible text |

Table 15: WCAG 2.0 — Language

| Rule ID | Impact | Description |
|---|---|---|
| html-has-lang | Serious | Document has a `lang` attribute |
| html-lang-valid | Serious | `lang` attribute on `<html>` has a valid value |
| html-xml-lang-mismatch | Moderate | `lang` and `xml:lang` agree on base language |
| valid-lang | Serious | `lang` attributes have valid values |

Table 16: WCAG 2.0 — Data Tables

| Rule ID | Impact | Description |
|---|---|---|
| td-headers-attr | Serious | Cells using `headers` refer only to cells in the same table |
| th-has-data-cells | Serious | `<th>` and header roles have data cells they describe |

Table 17: WCAG 2.0 — User Control & Timing

| Rule ID | Impact | Description |
|---|---|---|
| meta-refresh | Critical | `<meta http-equiv="refresh">` is not used for delayed refresh |
| no-autoplay-audio | Moderate | `<video>` or `<audio>` elements do not autoplay audio for more than 3 seconds without a control mechanism to stop or mute the audio |

Table 18: Best Practices — ARIA(Accessible Rich Internet Applications)

| Rule ID | Impact | Description |
|---|---|---|
| aria-allowed-role | Minor | `role` attribute has an appropriate value for the element |
| aria-dialog-name | Serious | ARIA dialog/alertdialog nodes have accessible names |
| aria-text | Serious | `role=text` used only on elements with no focusable descendants |
| aria-treeitem-name | Serious | ARIA treeitem nodes have accessible names |
| presentation-role-conflict | Minor | Presentational elements do not have global ARIA or `tabindex` |
| label-title-only | Serious | Every form element has a visible label and is not solely labeled using hidden labels, or the title or aria-describedby attributes |
| tabindex | Serious | `tabindex` attribute values are not greater than 0 |

Table 19: Best Practices — Landmarks & Regions

| Rule ID | Impact | Description |
|---|---|---|
| landmark-banner-is-top-level | Moderate | Banner landmark is top level |
| landmark-complementary-is-top-level | Moderate | Complementary/aside landmark is top level |
| landmark-contentinfo-is-top-level | Moderate | Contentinfo landmark is top level |
| landmark-main-is-top-level | Moderate | Main landmark is top level |
| landmark-no-duplicate-banner | Moderate | At most one banner landmark |
| landmark-no-duplicate-contentinfo | Moderate | At most one contentinfo landmark |
| landmark-no-duplicate-main | Moderate | At most one main landmark |
| landmark-one-main | Moderate | Document has a main landmark |
| landmark-unique | Moderate | Landmarks have unique role/name/title combinations |
| region | Moderate | All page content is contained by landmarks |
| skip-link | Moderate | All skip links have a focusable target |

Table 20: Best Practices — Headings & Structure

| Rule ID | Impact | Description |
|---|---|---|
| empty-heading | Minor | Headings have discernible text |
| heading-order | Moderate | Heading order is semantically correct |
| page-has-heading-one | Moderate | Page (or a frame) contains a level-one heading |
| empty-table-header | Minor | Table headers have discernible text |
| accesskeys | Serious | Every accesskey attribute value is unique |
| image-redundant-alt | Minor | Image alternative is not repeated as text |
| meta-viewport-large | Minor | `<meta name="viewport">` can scale a significant amount |

Table 21: Best Practices — Tables

| Rule ID | Impact | Description |
|---|---|---|
| scope-attr-valid | Moderate, Critical | `scope` attribute is used correctly on tables |
| table-duplicate-name | Minor | `<caption>` text differs from `summary` attribute |
| frame-tested | Critical | `<iframe>` and `<frame>` elements contain the axe-core script |

# G PROMPT DETAILS

We provide the details of the prompt used in our work.

## G.1 PROMPTS FOR DATASET SYNTHESIS

---

**Prompt for Generating Training Data**

```
Your task is to generate 100 different possible UI request
queries for a certain application domain category.

The application category is:
{category}
Requirements for queries:
1. Each query should include different page types possible
    within the app in the application domain
2. Each queries should be specific about widget requirements
    and context rich (side navigation, collapsible menus, etc.)
3. Each query must be different from the others
4. Queries should be realistic and natural like real user
    requests
5. Queries should be in some length and semantically rich,
    not just a few words
6. Queries should have solid use case or purpose, not random
    requests
7. Do not consider interactivity or animations or hovering
    effects focus on static UI elements
8. Queries require some ambience or style requests like
    color scheme, typography, etc.
```

---

**Prompt for Captioning metadata for evaluation set curation**

```
Caption given UI screenshot with
1. Main purpose and intent of the UI, regarding the target
   audience and actual use case.
2. Page type of the UI, such as a landing page, blog post,
   product page, etc. (in short-answer format)
3. Domain of the UI, such as e-commerce, social media,
   education, etc. (in short- answer format)
4. Top 5 important visual elements in the UI design, that are
   crucial for more user engagement and usability. But you
   should EXCLUDE any elements about the EXACT IMAGES in the
   UI.
Answer in such format:
1. <Purpose/Intent>
2. <Page Type>
3. <Domain>
4. (a) <Element (a)>
(b) <Element (b)>
(c) <Element (c)>
...
```

---

**Prompt for generating RealUIReq-300 evaluation set queries**

```
SYSTEM PROMPT:
You are a helpful assistant that generates realistic user
requests for web UI development.

USER PROMPT:
Based on the following web page specifications, generate a
user request that mentions EVERY DETAIL provided, including
the purpose, page type, domain, and all listed components.

Purpose: {purpose}
Page Type: {page_type}
Domain: {application_domain}
Required Components: {required_components}

The request should be 3-5 sentences long and sound realistic.
```

## G.2  PROMPTS FOR INFERENCE AND EVALUATION

---

**Prompt for Inference of models for web UI generation**

```
You are an expert UI designer assistant.
You should plan the design based on the user request.
Show the plan in the `<think>` tag.
    - You must think about the html structure and widgets
      needed to fulfill the user request.
    - You must think about the Tailwind CSS classes to use
      for styling.
Then, you should generate a complete HTML document that
includes:
    - A `<head>` section with a `<meta charset="UTF-8">` tag
    - A `<meta name="viewport" content="width=device-width,
      initial-scale=1.0">` tag
    - A proper tailwind css link tag to load Tailwind CSS
      from CDN
    - A `<body>` section that contains the complete HTML
      structure and content
The HTML document should be visually appealing, well-
structured, and content/semantically-rich.
You must strictly follow the output format shown below:
<think>
...
</think>

<answer>
```html
<html>
<head>
    <meta charset="UTF-8" />
    <meta name="viewport" content="width=device-width,
    initial-scale=1.0" />
    ...
</head>
<body>
...
</body>
</html>
```
</answer>

User: {user_request}
Assistant:
```

---

**WebGen-Bench Appearance Score Evaluation Prompt (Lu et al., 2025)**

```
Instruction: You are tasked with evaluating the functional
design of a webpage that had been constructed based on the
following instruction: {instruction}

Grade the webpage's appearance on a scale of 1 to 5 (5 being
highest), considering the following criteria:
    - Successful Rendering: Does the webpage render correctly
      without visual errors? Are colors, fonts, and components
      displayed as specified?
    - Content Relevance: Does the design align with the
      webpage's purpose and user requirements? Are elements
      (e.g., search bars, report formats) logically placed and
      functional?
    - Layout Harmony: Is the arrangement of components (text,
      images, buttons) balanced, intuitive, and clutter-free?
    - Modernness & Beauty: Does the design follow contemporary
      trends (e.g., minimalism, responsive layouts)? Are
      colors, typography, and visual hierarchy aesthetically
      pleasing?
Grading Scale:
    - 1 (Poor): Major rendering issues (e.g., broken layouts,
      incorrect colors). Content is irrelevant or missing.
      Layout is chaotic. Design is outdated or visually
      unappealing.
    - 2 (Below Average): Partial rendering with noticeable
      errors. Content is partially relevant but poorly
      organized. Layout lacks consistency. Design is basic
      or uninspired.
    - 3 (Average): Mostly rendered correctly with minor flaws.
      Content is relevant but lacks polish. Layout is
      functional but unremarkable. Design is clean but lacks
      modern flair.
    - 4 (Good): Rendered well with no major errors. Content is
      relevant and logically organized. Layout is harmonious
      and user-friendly. Design is modern and visually
      appealing.
    - 5 (Excellent): Flawless rendering. Content is highly
      relevant, intuitive, and tailored to user needs. Layout
      is polished, responsive, and innovative. Design is
      cutting-edge, beautiful, and memorable.

Task: Review the provided screenshot(s) of the webpage.
Provide a detailed analysis and then assign a grade (from 1
to 5) based on your analysis. Highlight strengths, weaknesses,
and how well the design adheres to the specifications, but
don't mind the absence of images or cards for specific data
because they are not the target for evaluation.

IMPORTANT: Please end your response with a clear grade in the
format "Grade: X" where X is a number from 1 to 5.

Your Response Format:
  Analysis: [from 2 to 4 paragraphs addressing all criteria,
            referencing the instruction]
  Grade: [from 1 to 5]
  Your Response:
```

## H  QUALITATIVE WEB UI STYLE DIVERSITY OF A11YN

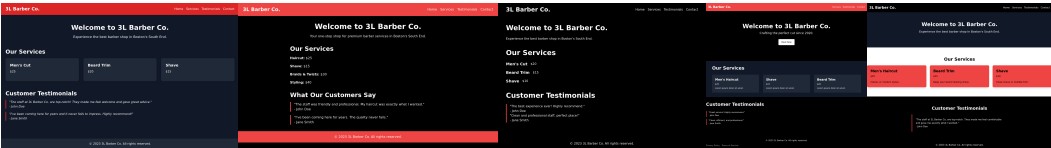

Figure 8: Style Diversity of A11yn for the UI Generation Prompt: *I want a landing page for 3L Barber Co., a barbershop located in Boston's South End. The page should attract and inform potential customers about our services and atmosphere, featuring a bold, contrasting color scheme of black, white, and red. I'd like to see large, eye-catching typography for the main headline, along with a clear navigation menu that includes essential sections. Additionally, please include a detailed list of services with prices and a section for customer testimonials and ratings.*

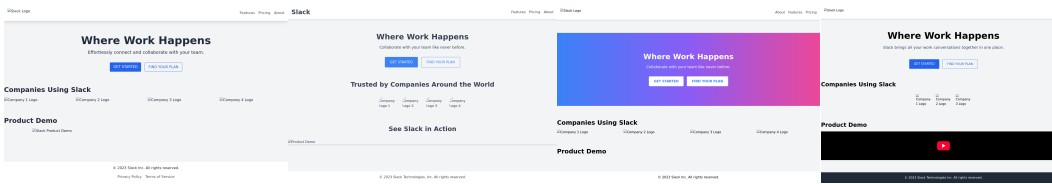

Figure 9: Style Diversity of A11yn for the UI Generation Prompt: *Please make a landing page designed to promote Slack as a collaborative workspace platform for teams. The page should feature a bold, attention-grabbing headline that says "Where Work Happens," along with prominent "GET STARTED" and "FIND YOUR PLAN" call-to-action buttons. Additionally, please include logos of well-known companies that use Slack, a clean and minimalist design with ample white space, and an interactive product demo or screenshot that showcases key features. The domain should focus on business communication and collaboration.*

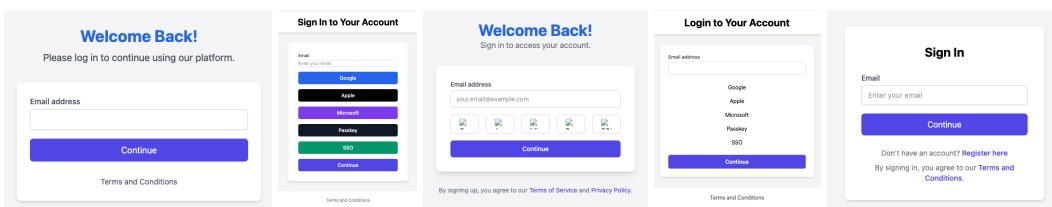

Figure 10: Style Diversity of A11yn for the UI Generation Prompt: *I would like to have a login page for our productivity and collaboration platform that provides a sign-in interface for users to access membership content. This page should include multiple authentication options such as Google, Apple, Microsoft, passkey, and SSO for convenience and security. Additionally, please ensure there is a clear and concise page title explaining the purpose, an email input field for manual login, a large, prominent "Continue" button, and a link to the terms and conditions for transparency.*

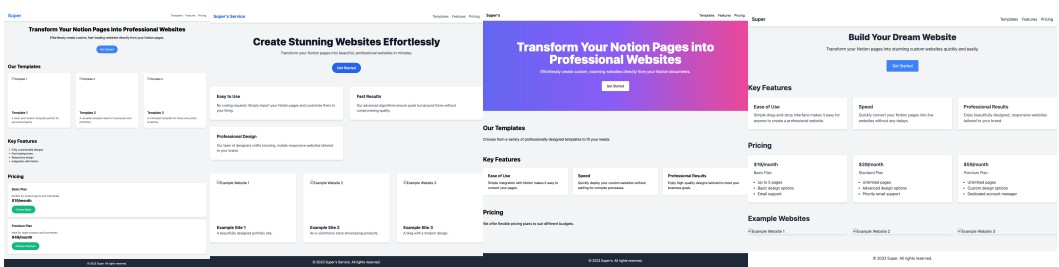

Figure 11: Style Diversity of A11yn for the UI Generation Prompt: *I want a landing page for Super's service that creates custom websites from Notion pages. The purpose of the page is to attract and inform potential users about the ease of use, speed, and professional results of our service. It should include a clear, prominent headline explaining our core value proposition, a concise subheading detailing key benefits, and a prominent call-to-action button to get started. Additionally, please include a navigation menu with sections like Templates, Features, and Pricing, along with example website previews to showcase our capabilities. The domain is web development/SaaS.*

