# OpenReview forum: "A11YN: Aligning LLMs for Accessible Web UI Code Generation"
_ICLR.cc/2026/Conference — ICLR 2026 Conference Withdrawn Submission_

### Official Review · Reviewer_pBvL · 2025-11-01

**Soundness:** 2
**Presentation:** 3
**Contribution:** 2
**Rating:** 4
**Confidence:** 3

**Summary:**

The paper introduces A11YN, an approach to align large language models for generating more accessible web UI code. The method integrates accessibility evaluation (via Axe-core) into a GRPO-based reinforcement learning framework and fine-tunes an open-source code model using synthetic accessibility-focused datasets (UIReq-6.8K and RealUIReq-300). Experiments show reduced violation rates and improved accessibility scores compared with baselines.

**Strengths:**

+ Socially meaningful objective. Addressing accessibility in code generation is an important and underexplored direction that broadens the scope of model alignment beyond general coding accuracy.

+ Clear pipeline design. The integration of automatic accessibility evaluation into a reinforcement learning loop is well described and straightforward to reproduce.

+ Comprehensive baseline coverage. The experiments include multiple strong models (e.g., GPT-4, CodeLlama, DeepSeek-Coder), providing a fair empirical comparison within the same evaluation setup.

**Weaknesses:**

- Limited methodological novelty.  The paper mainly applies an existing RL-based alignment framework (GRPO) to web accessibility without introducing new algorithmic ideas or training mechanisms. While the topic is valuable, the technical contribution is largely an adaptation rather than a conceptual advance.


- Synthetic and weakly validated data.  Both the training dataset (UIReq-6.8K) and the benchmark (RealUIReq-300) rely heavily on GPT-generated instructions and code. This raises concerns about data authenticity and generalization, as the method is essentially evaluated on distributions created by the same type of model.


- Evaluation lacks robustness.  The reported improvements mainly rely on Axe-core–based automatic scores. There is no human or cross-tool validation, nor analysis of code correctness or executability after alignment. The gains therefore reflect optimization toward the reward function rather than clear practical improvements in accessibility.

**Questions:**

1. How well does the proposed method generalize to real-world accessibility requests beyond GPT-generated synthetic data?


2. Can the authors provide human or cross-tool evaluations to confirm that the observed improvements go beyond optimizing for the Axe-core reward?

---

> ### Author Response · Authors · 2025-11-22
>
> Dear Reviewer pBvL
>
> Thank you for your thoughtful assessment. We have addressed your concerns below.
>
> # W1: Methodological Novelty
>
> We agree that our work does not propose a new technical RL methodology. Instead, our contribution lies in making RL-based alignment feasible for web accessibility, a domain where collecting large-scale accessible code data for supervised finetuning or preference learning is impractical.
>
> While GRPO is known to perform well in domains with fully verifiable, deterministic rewards (e.g., mathematics), its applicability to non-verifiable, human-centered tasks such as accessible UI code generation has remained unclear. Our work shows that **accessibility can be transformed into a verifiable alignment target**, enabling RL to operate meaningfully where *human feedback or preference data is unavailable*.
>
> The design and validation of this accessibility reward therefore form the core of our technical contribution. It resolves the key barrier that previously prevented RL-based alignment in this space and makes **RL viable in a socially important but data-scarce domain, such as web accessibility**.
>
>
> # W2, Q1: Prompt Data Synthesis
>
> While we acknowledge the limits of machine-assisted data generation, we took explicit steps to keep both UIReq-6.8K and RealUIReq-300 realistic, diverse, and suitable for evaluating generalization.
>
> ## 1. Real-application grounded prompt design
>
> Both UIReq-6.8K and RealUIReq-300 are generated under task protocols tied to concrete application domains (Appendix A), rather than free-form GPT text. As described in Appendix F.1, we prompt for precise, function-level requests that must include specific UI elements (for example, side navigation, collapsible menus, concrete components from our taxonomy) instead of vague layout descriptions. This constrains GPT to produce prompts that correspond to implementable UI behaviors rather than abstract website descriptions.
>
> ## 2. Inverse generation from real websites and human auditing
>
> RealUIReq-300 is constructed by inverse-generating accessibility-oriented requests from real websites and applications, not from arbitrary synthetic scenarios. Each evaluation prompt is then manually audited to (1) ensure that it correctly reflects the source interface, (2) remove vague or underspecified requests, and (3) maintain coverage over diverse layouts and interaction patterns (Lines 269–284). In this sense, GPT is used as a drafting tool, while the final benchmark distribution is anchored in real interfaces and filtered by humans.
>
>
> # W3, Q2: Cross Validation for Evaluation Robustness
>
> Thank you for raising the concern about robustness. We agree with the need for additional validation to ensure that the improvements go beyond Axe-core–specific optimization. However, following conventions in prior HCI and accessibility research, human accessibility evaluation is known to be limited, inconsistent, and difficult to reproduce, and prior work explicitly reports that “automated software tools have the ability to discover more accessibility errors than human experts” [1]. Because of this, automated accessibility tools are already the dominant and widely accepted evaluation metric in the literature, used in most published accessibility assessment studies.
> Regarding correctness and executability, Web UI code correctness is determined by whether the rendered UI from code semantically matches the natural language request, which we evaluate through the appearance score from WebGenBench. Executability is inherently guaranteed—every generated HTML/CSS sample is successfully rendered into a concrete UI.
>
> To directly address your request, we additionally conducted a cross-tool validation using AChecker[3], a widely used accessibility auditor also adopted in recent work [2]. The AChecker results closely mirror the Axe-core trend: A11yn shows enhanced performance over the Base model (Qwen2.5-Coder-7B-Instruct) and consistently produces substantially fewer accessibility violations than FeedA11y, demonstrating the generalization across independent auditing frameworks.
>
> | Model      | Total Violations (↓) | Avg. Violations per page (↓) |
> |------------|------------------|-------------------------|
> | Qwen2.5-Coder-7B-Instruct (Base)| 2,891        | 9.64     |
> | FeedA11y   | 2,576            | 8.59                    |
> | **A11yn**  | **1,347**        | **4.49**                |
>
> [1] Alsaeedi, Abdullah. "Comparing web accessibility evaluation tools and evaluating the accessibility of webpages: proposed frameworks." Information 11.1 (2020): 40.
>
> [2] Suh, Hyunjae, et al. "Human or LLM? A Comparative Study on Accessible Code Generation Capability." arXiv preprint arXiv:2503.15885 (2025).
>
> [3] https://achecks.org/achecker/

---

### Official Review · Reviewer_Zw7A · 2025-11-02

**Soundness:** 3
**Presentation:** 3
**Contribution:** 2
**Rating:** 4
**Confidence:** 3

**Summary:**

This work proposes to train website coding LLM with a reward function that comes directly from an automated WCAG auditor (axe‑core), which measures the violations of accessibility defects in HTML/CSS. For training prompts, the authors synthesize UI requests with GPT‑4o‑mini to form UIReq‑6.8K spanning 68 application categories. For evaluation, they collect screenshots of public webpages, extract structured metadata, and prompt another GPT model to generate user requests from the metadata. The LLM coder is GRPO‑tuned against the accessibility reward. The models are scored by WCAG violations and the appearance of the generated website. The results show that the proposed method improves accessibility without degrading visual/semantic quality.

**Strengths:**

1. This work presents a novel application of GRPO for improving accessibility of generated websites.

2. The newly curated dataset may be useful for future work on website coding.

3. The proposed approach is simple and effective.

**Weaknesses:**

1. The technical depth is limited: Although the application of GRPO for improving accessibility of website coding is straightforward but somewhat incremental, training LLMs with hand‑engineered rewards is already well explored.

2. Using only one testset leaves generalization to unseen domains and request styles, especially those not represented during training, unclear.

3. Both training and eval prompts are GPT‑generated, and it's unclear that the synthesized requests are realistic. As seen in Figure 4, the many remain high‑level and omit concrete functional elements (e.g., dropdowns, buttons).

**Questions:**

1. Whether the RL training decreases the diversity of website generation? The code LLM may learn a specific UI style, which is a safe solution.

---

> ### Author Response · Authors · 2025-11-22
>
> Dear Reviewer Zw7A,
>
> Thank you for sharing your concerns. We have addressed your concerns below.
>
> # W1: Technical Depth
>
> Our technical contribution lies in making RL-based alignment feasible for web accessibility, a domain where collecting large-scale accessible code data for supervised finetuning or preference learning is impractical.
>
> While GRPO is known to perform well in domains with fully verifiable, deterministic rewards (e.g., mathematics), its applicability to non-verifiable, human-centered tasks such as accessible UI code generation has remained unclear. Our work shows that **accessibility can be transformed into a verifiable alignment target**, enabling RL to operate meaningfully where *human feedback or preference data is unavailable*.
>
> The design and validation of this accessibility reward therefore form the core of our contribution. It resolves the key barrier that previously prevented RL-based alignment in this space and makes **RL viable in a socially important but data-scarce domain, such as web accessibility**.
>
> # W2: RealUIReq-300 test set as part of our main contribution
>
> We agree that generalization is an important concern. In our setting, the key challenge was the lack of an academically open benchmark for natural-language–to–web-UI generation that reflects realistic, multi-sentence requests with explicit UI intent and component requirements.
>
> While Screen2Words [1] is a well-known dataset, it was explicitly designed for *UI summarization*, not UI generation; its annotations provide short taxonomic descriptions (like `sign in page`) of Android screens rather than request-style inputs that specify *purpose*, *page type*, *domain*, and *required UI components*. This distinction is crucial, because in NL to UI code generation, these elements act as **UI-instance–specific evaluation criteria**, which define what the model is expected to build, what purpose the web UI must serve, and which UI components (e.g. Button, Navigation Bar) must appear. Without such semantic anchors, it is challenging to evaluate whether a generated UI meaningfully satisfies the user request. As such, Screen2Words could not serve as a reliable benchmark.
>
> The training data was designed to avoid overfitting to a narrow range of domains. UIReq-6.8K spans 68 diverse application domains (`Appendix A`), guiding the model to learn domain-general UI construction behaviors rather than memorizing templates tied to specific categories.
>
> RealUIReq-300 introduces deliberate distribution shift, drawing from roughly 61 distinct real-world use-case domains across commerce, education, nonprofit, media, business services, and more. Because these pages were independently crawled and then converted into multi-sentence requests with human refinement, they naturally exhibit unseen phrasing styles, UI intents, and layout structure requirements relative to UIReq-6.8K. We **included the full domain breakdown** of RealUIReq-300 in `Appendix B`  for further clarity.
>
> Finally, our evaluation follows emerging practice in UI generation, where criteria-driven MLLM-based judges (e.g., WebGenBench[2]) are increasingly used to assess appearance and semantic fidelity. This complements our accessibility-focused metrics and provides a broader perspective on model generalization.
>
> [1] Wang, Bryan, et al. "Screen2words: Automatic mobile ui summarization with multimodal learning." The 34th Annual ACM Symposium on User Interface Software and Technology. 2021
>
> [2] Lu, Zimu, et al. "WebGen-Bench: Evaluating LLMs on Generating Interactive and Functional Websites from Scratch." arXiv preprint arXiv:2505.03733 (2025).

---

> ### Author Response · Authors · 2025-11-22
>
> # W3: Prompt Data Synthesis
>
> While we acknowledge the limit of machine-assisted annotations, we emphasize that extensive measures were taken to preserve the quality and diversity of our datasets.
>
> 1. *We prompt for precise, function-level requests rather than high-level descriptions.*
> `Appendix F.1` details our criteria, which require explicit functional elements such as side navigation, collapsible menus, action buttons, or tables. The prompts are constructed to specify concrete UI behaviors rather than abstract layout intentions.
>
> 2. *We manually filter and audit the entire evaluation set.*
> Evaluation prompts are inverse-generated from real websites, and each item is reviewed by annotators to ensure specificity, validity, and alignment with our component taxonomy `(Lines 269–284)`. This process removes vague or underspecified queries that could introduce noise.
>
> 3. *Figure 4 illustrates the data generation workflow and is not intended to represent full request contents.*
> The prompts shown in the figure are abbreviated for conceptual clarity. A representative evaluation prompt is provided below for context:
>
> `“Generate a landing page designed for an e-commerce website that aims to attract and inspire entrepreneurs to start their business using Shopify's platform. The page should feature a large, bold headline text that captures attention, along with a concise subheading that explains the value proposition. Additionally, please include clear call-to-action buttons for ‘Start for free’ and ‘View plans,’ the Shopify logo for brand recognition, and a menu bar for easy navigation. The overall goal is to emphasize the growth potential and ease of use of Shopify for new entrepreneurs.”`

---

> ### Author Response · Authors · 2025-11-22
>
> # Q1: Impact on Diversity of UI style generation
>
> We agree that RL could, in principle, has a possibility of guiding model towards a “safe” and stylistically narrow solution. To check this, we examined stylistic variation by sampling multiple generations (n=5 with temperature 0.7) for the same instruction of the test set. Across these samples, A11yn consistently produced distinct layouts and design structures, indicating that RL did not collapse the model to a certain UI style.
>
> While qualitative style judgments are inherently subjective, inspection among the authors found *clear and diverse design style patterns* in A11yn’s outputs. To make this transparent, we **included several UI samples in the `Appendix H`** about several instructions from our evaluation set with multiple rendered samples per instruction, allowing visual assessment of the UI style diversity.

---

### Official Review · Reviewer_sosE · 2025-11-02

**Soundness:** 3
**Presentation:** 3
**Contribution:** 3
**Rating:** 4
**Confidence:** 3

**Summary:**

This paper presents a new method to align code generation LLMs to produce accessibility-compliant web UIs. The approach augments existing LLMs with a new reinforcement learning reward. The reward takes into account violations of accessibility as negative rewards and trains the LLM using GRPO. To evaluate the performance, the authors also curated a new benchmark consisting 300 real-world web UIs. Results show that the propose method outperforms the baselines including Claude Sonnet 4, reducing the inaccessibility rate by 60% over the base model, while preserving semantic fidelity and visual quality of generated UIs.

**Strengths:**

1.	The paper concerns accessibility as a new metric for text-to-UI generation, which is an important research aspect.
2.	The paper is generally well-structured. The motivation, methodology, and results are easy to follow.
3.	The authors provide a new benchmark, RealUIReq-300, containing realistic web UI requests curated from 300 real-world web pages, which can be useful for future evaluation.

**Weaknesses:**

1.	The core methodological novelty is modest. The proposed method merely augments GRPO with a custom accessibility reward. While practically useful, this design does not introduce fundamentally new RL methodology or optimization mechanisms, making the novelty borderline for ICLR standards.
2.	The qualitative study in 6.2 is rather shallow. It provides only a single qualitative example (color contrast). A deeper case analysis, covering diverse violation types (e.g., ARIA roles, landmark semantics, keyboard navigation), would strengthen the narrative.
3.	The paper evaluates accessibility solely via automated tools (Axe-core). Given the human-centered nature of accessibility, even a small-scale user or expert study would greatly enhance credibility.
4.	There is no ablation showing the contribution of different reward components or GRPO hyperparameters. It is unclear how sensitive the results are to the weighting scheme, the base score B, or the severity mapping.
5.	The work focuses narrowly on HTML-based UIs. The paper could discuss whether the proposed alignment strategy generalizes to other UI platforms (e.g., React, Flutter, mobile UIs) or broader code-generation tasks.
6.  There are also some recent related work on Web UI code generation, which can be discussed. For example:

UICopilot: Automating UI Synthesis via Hierarchical Code Generation from Webpage Designs, https://arxiv.org/abs/2505.09904

Unlocking the conversion of web screenshots into html code with the websight dataset. 2024. URL https://api.semanticscholar.org/CorpusID:268385510.

VISION2UI: A Real-World Dataset with Layout for Code Generation from UI Designs, https://arxiv.org/abs/2404.06369v1, April 2024.

**Questions:**

- Did you perform a user or expert study on accessibility?
- Any ablation study showing the contribution of different reward components?

---

> ### Author Response · Authors · 2025-11-22
>
> Dear Reviewer sosE
>
> We truly appreciate your insights and time for reviewing our work. We have addressed your concerns below.
>
> # W1: Methodological Novelty
>
> We agree that our work does not propose a new RL methodology. Instead, our contribution lies in making RL-based alignment feasible for web accessibility, a domain where collecting large-scale accessible code data for supervised finetuning or preference learning is impractical.
>
> While GRPO is known to perform well in domains with fully verifiable, deterministic rewards (e.g. mathematics), its applicability to non-verifiable, human-centered tasks such as accessible web UI code generation has remained unclear. Our work shows that **accessibility can be transformed into a verifiable alignment target**, enabling RL to operate meaningfully where *human feedback or preference data is unavailable*.
>
> The design and validation of this accessibility reward therefore form the core of our contribution. It resolves the key barrier that previously prevented RL-based alignment in this space and makes **RL viable in a socially important but data-scarce domain, such as web accessibility**.
>
> # W2: Deeper Case Analysis for `6.2`
>
> Thank you for the feedback. We agree that an alternative sample pair with broader qualitative analysis in `6.2` would strengthen the section. We **included another Base Model and A11yn sample pair with accessibility tree comparison** that represent a wider range of accessibility issues, which entails `color contrast`, `accessible DOM name`, and `landmark semantics`  to provide a more comprehensive view of how our method improves multiple violation types beyond color contrast.

---

> ### Author Response · Authors · 2025-11-22
>
> # W3, Q1: Additional Validation for Credibility
>
> We acknowledge the need of human study considering the human-centered nature of accessibility. However, our decision of using accessibility tools reflects both well-documented constraints in accessibility research and established methodological practice in HCI.
>
> First, conducting user studies with disabled participants is inherently challenging. A recent large-scale survey of CHI and ASSETS work [1] highlights *persistent recruitment difficulties* and the *limited availability of participants from many disability groups*. Prior work [4] also shows that expert or user-driven accessibility evaluations are time-intensive and inconsistent: evaluating a single webpage can require up to an hour of focused attention, and attention degradation across repeated tasks reduces reliability. Work like [8] explicitly mentions that *“automated software tools have the ability to discover more accessibility errors than human experts”*. Such constraints make expert or small-scale human evaluations impractical for large-scale, model-level benchmarking.
>
> Second, due to such challenges, many HCI accessibility studies rely on WCAG-based automated tools for assessment or enhancement [2,6]. Automated auditors provide deterministic, consistent judgments, enabling systematic evaluation at scales that are infeasible with human studies. Among these tools, Axe-Core [3] is widely adopted and built on extensive WCAG expertise, benefiting from over 4,500 contributions from Deque and the broader accessibility community. Its rule coverage and reliability have also been examined in HCI research [4].
>
> We acknowledge that using a single audit tool may introduce coverage limitations. Prior HCI work shows that cross-tool validation improves robustness by capturing complementary subsets of WCAG criteria [2,5]. Following this guidance, we additionally evaluate using AChecker, a widely used auditor also employed in the web accessibility field [2]. The AChecker results parallel the Axe-core findings: A11yn shows enhancement from its base model (Qwen2.5-Coder-7B-Instruct), and more reduced violations compared to FeedA11y, demonstrating that our improvements generalize across multiple auditing tools.
> | Model | Total Violations (↓) | Avg. Violations per page (↓) |
> |------|--------|----|
> | Qwen2.5-Coder-7B-Instruct (Base) | 2,891|9.64|
> | FeedA11y| 2,576| 8.59|
> | **A11yn**| **1,347** | **4.49**|
>
> [1] Kelly Mack, Emma McDonnell, Dhruv Jain, Lucy Lu Wang, Jon E. Froehlich, and Leah Findlater. 2021. What Do We Mean by "Accessibility Research"?: A Literature Survey of Accessibility Papers in CHI and ASSETS from 1994 to 2019. Proceedings of the 2021 CHI Conference on Human Factors in Computing Systems. Association for Computing
>
> [2] Suh, Hyunjae, et al. "Human or LLM? A Comparative Study on Accessible Code Generation Capability." arXiv preprint arXiv:2503.15885 (2025).
>
> [3] https://www.deque.com/axe/axe-core/
>
> [4] Fischer, T., Lundell, B. & Gamalielsson, J. Coverage of web accessibility guidelines provided by automated checking tools. Univ Access Inf Soc (2025).
>
> [5] Pool, Jonathan Robert. "Accessibility metatesting: comparing nine testing tools." Proceedings of the 20th International Web for All Conference. 2023.
>
> [6] Mowar, Peya, et al. "CodeA11y: Making AI Coding Assistants Useful for Accessible Web Development." Proceedings of the 2025 CHI Conference on Human Factors in Computing Systems. 2025.
>
> [7] https://achecks.org/achecker/
>
> [8] Alsaeedi, Abdullah. "Comparing web accessibility evaluation tools and evaluating the accessibility of webpages: proposed frameworks." Information 11.1 (2020): 40.

---

> ### Author Response · Authors · 2025-11-22
>
> # W4, Q2: Ablation Study for Accessibility Reward
>
> Thank you for your point. To assess the improvements from the proposed **severity-weighted reward**, we conduct an ablation comparing **weighted** and **uniform** reward setting. The weighted configuration assigns penalties of (0.1, 0.2, 0.3, 0.4) to Minor, Moderate, Serious, and Critical violations respectively, whereas the uniform variant applies equal weights of (0.25, 0.25, 0.25, 0.25). Candidates were evaluated using greedy decoding and assessed with two independent accessibility auditors—Axe-core, which provides severity-level breakdowns, and AChecker, which reports overall violation counts and inaccessibility rates.
>
> | Model      | Minor (Axe-Core) (↓) | Moderate (Axe-Core) (↓) | Serious (Axe-Core) (↓) | Critical (Axe-Core) (↓) | Violation Count (AChecker) (↓) | Inaccessibility Rate (AChecker) (↓) |
> |------------|-------|----------|---------|----------|----------------|---------------------------------|
> | Base    |   0   |   1130   |   786   |   46   | 2891 | 0.20 |
> | A11yn (uniform) | 3 | 422 | 474 | 30 | 1618 | 0.12 |
> | A11yn (weighted) | **0** | **244** | **472** | **23** | **1324** | **0.10** |
>
> The results shows that the **weighted** reward consistently outperforms the uniform reward across all four Axe-core severities and yields the lowest violation count under AChecker. Notably, the weighted model demonstrates evident reduction in violations especially: *Moderate (↓78%)* and *Critical (↓50%)*.
>
> This pattern reflects two effects observed in accessibility evaluation. First, the weighted reward places strong optimization pressure on Critical violations due to their higher penalty, encouraging the model to correct major structural issues. Second, because Moderate violations are both highly frequent and often co-occur with the higher-severity defects, fixing the underlying structural problems resolves many Moderate violations as a downstream effect. In contrast, the uniform reward distributes penalty evenly across heterogeneous violation types, leading to fewer of these structural corrections. Thus, empirical trend proves that severity weighting leads to broader and more robust accessibility improvements.
>
> # W5: Narrow Scope of UI
>
> We agree that a production-ready system should ideally operate across diverse UI frameworks. However, our scope is intentional: verifiable RL-based alignment requires **(1) a deterministic, reproducible code generation and rendering pipeline** and **(2) a stable, expert-maintained accessibility oracle**. HTML/CSS was a suitable environment that robustly satisfied both conditions.
> Focusing on HTML/CSS is not a narrow design choice. HTML/DOM serves as the execution substrate for the majority of modern UI frameworks like React, Next.js, Flutter web, ultimately compiling down to DOM structures with HTML-compatible accessibility semantics. While frameworks differ at the component-syntax level, their accessibility behavior is uniformly evaluated on the DOM. This makes HTML a unifying target rather than a framework-specific one. Recent UI-generation benchmarks such as Design2Code [2] adopt HTML/CSS as a typical target for front-end web UI representation.
> At the same time, our goal is not to design a universally platform-agnostic pipeline, but to study how verifiable accessibility rewards affect alignment dynamics on top of existing web UI generation capabilities. This scoped methodology is consistent with prior UI-generation research: for instance, UICoder [1] focuses exclusively on SwiftUI to study automated feedback-driven code improvement. Similarly, anchoring in HTML/CSS allows us to isolate the impact of the accessibility reward without introducing variability from differing framework syntaxes or immature component-level accessibility tooling. It also aligns with the coder-model practice, where HTML/CSS remains the most reliable and well-supported domain for LLM-based web UI generation [3].
> For reproducibility and consistency with established methodology in UI-generation research [1,2], we therefore conduct our experiments in HTML/CSS. Extending verifiable accessibility alignment to multi-environments is a promising future direction, and our approach provides a foundation for enabling such work once stable framework-level accessibility oracles become available.
>
> [1] Wu, J., Schoop, E., Leung, A., Barik, T., Bigham, J., & Nichols, J. (2024). UICoder: Finetuning Large Language Models to Generate User Interface Code through Automated Feedback. In Proceedings of NAACL-HLT 2024 (Vol. 1, pp. 7511–7525). Association for Computational Linguistics.
>
> [2] Si, C., Zhang, Y., Li, R., Yang, Z., Liu, R., & Yang, D. (2025). Design2Code: Benchmarking Multimodal Code Generation for Automated Front-End Engineering. In Proceedings of NAACL-HLT 2025 (Vol. 1, pp. 3956–3974). Association for Computational Linguistics
>
> [3] Hui, Binyuan, et al. "Qwen2. 5-coder technical report." arXiv preprint arXiv:2409.12186 (2024).

---

> ### Author Response · Authors · 2025-11-22
>
> # W6: Consideration of Image to UI Code Generation
>
> We appreciate the additional related works. These works, however, focus on **vision-based UI code generation (screenshot/UI design to code)**, whereas our work centers on the **Natural Language to Web UI generation** setting. Because the input modality and the scope of generation task differs, we did not position them as core comparisons, but we will clarify this distinction in the revision.

---

### Author Response · Authors · 2025-11-22
**General Response**

We appreciate all reviewers for their thoughtful and constructive feedback.

Reviewers consistently noted the following strengths:
- **Meaningful and Novel Research Direction**
    - Introducing **accessibility** as a core evaluation and optimization metric for text-to-UI generation.
    - Tackling a **socially impactful** but **underexplored** problem beyond traditional code correctness.
- **Clear and Reproducible Methodology**
    - A well-motivated problem framing supported by a **clear and coherent pipeline design**.
    - A methodology that is **straightforward, transparent, and reproducible**.
- **Useful Dataset Contribution**
    - Release of **RealUIReq-300**, a collection of realistic UI requests from 300 real-world websites.
    - Recognized as a valuable benchmark for future research on UI code generation and accessibility evaluation.

We have carefully addressed each reviewer’s comments in detail. Please feel free to share any additional feedback or concerns for further discussion.

---

### Note · Authors · 2025-12-03

I have read and agree with the venue's withdrawal policy on behalf of myself and my co-authors.